Anopheles gambiae densovirus (AgDNV) has negligible effects on adult survival and transcriptome of its mosquito host

Ren Xiaoxia 2
Hughes Grant L. 1
Niu Guodong 1 4
Suzuki Yasutsugu 1
Rasgon Jason L. 1 jlr54@psu.edu
1 The Department of Entomology, Center for Infectious Disease Dynamics, and the Huck Institutes of the Life Sciences, Pennsylvania State University , University Park, PA , USA
2 Pharmaceutics International Inc. , Hunt Valley, MD , USA
Newton Irene
4 Current affiliation: Department of Chemistry and Biochemistry, University of Oklahoma, Norman, OK, USA

Electronic publication date: 2014 Sep 18
Publication date: 2014
Volume: 2
Electronic Location ID: e584
Received 2014 Aug 7; Accepted 2014 Aug 28
Copyright: © 2014 Ren et al.
Copyright year: 2014
Copyright holder: Ren et al.
License: This is an open access article distributed under the terms of the Creative Commons Attribution License, which permits unrestricted use, distribution, reproduction and adaptation in any medium and for any purpose provided that it is properly attributed. For attribution, the original author(s), title, publication source (PeerJ) and either DOI or URL of the article must be cited.
License URL: https://creativecommons.org/licenses/by/4.0/

Keywords: Anopheles, Densovirus, Paratransgenesis, Fitness, Mosquito, Transcriptome, Malaria

Funding: NIH R21AI088311 R21AI111175 R21AI070178 This research was supported by NIH grants R21AI088311, R21AI111175 and R21AI070178 to JLR. The funders had no role in study design, data collection and analysis, decision to publish, or preparation of the manuscript.

==============================
Mosquito densoviruses (DNVs) are candidate agents for paratransgenic control of malaria and other vector-borne diseases. Unlike other mosquito DNVs, the Anopheles gambiae DNV (AgDNV) is non-pathogenic to larval mosquitoes. However, the cost of infection upon adults and the molecular mechanisms underpinning infection in the mosquito host are unknown. Using life table analysis, we show that AgDNV infection has minimal effects on An. gambiae survival (no significant effect in 2 replicates and a slight 2 day survival decrease in the third replicate). Using microarrays, we show that AgDNV has very minimal effect on the adult mosquito transcriptome, with only 4–15 genes differentially regulated depending on the statistical criteria imposed. The minimal impact upon global transcription provides some mechanistic understanding of lack of virus pathogenicity, suggesting a long co-evolutionary history that has shifted towards avirulence. From an applied standpoint, lack of strong induced fitness costs makes AgDNV an attractive agent for paratransgenic malaria control.

Introduction

Human malaria infects up to 500 million people per year and causes almost 3 million deaths annually (Hay et al., 2004). Traditional control strategies that target the mosquito vector (such as insecticides) are becoming less effective due to the emergence of resistance (Enayati & Hemingway, 2010). Therefore, there is a concerted effort to develop novel strategies to combat arthropod-borne diseases. One such strategy is to use microbes to manipulate components of host vector competence. While some microbes can either directly or indirectly affect Plasmodium development in the host (Cirimotich et al., 2011; Ricci et al., 2011; Hughes et al., 2011a; Hughes et al., 2014), the genetic modification of mosquito symbiotic microorganisms with effector molecules which inhibit pathogens (paratransgenesis) has been proposed as one novel method to control malaria (Favia et al., 2007; Wang et al., 2012). To be a microbial candidate for paratransgenic malaria control, the microorganism should not significantly compromise the host fitness and must be manipulatable to produce effector molecules of interest (Beard, Cordon-Rosales & Durvasula, 2002; Durvasula et al., 2003).

Densonucleosis viruses (or densoviruses (DNVs)) are icosahedral, non-enveloped parvoviruses that have been identified from many invertebrate taxa, including multiple mosquito species (Boublik, Jousset & Bergoin, 1994; Jousset, Baquerizo & Bergoin, 2000; Ledermann et al., 2004; Carlson, Suchman & Buchatsky, 2006; Ren, Hoiczyk & Rasgon, 2008; Zhai et al., 2008; Ng et al., 2011). DNVs are easily to manipulate and are candidate agents for paratransgenic control of vector-borne diseases by expression of toxins or anti-pathogen effector molecules (Ren, Hoiczyk & Rasgon, 2008; Suzuki et al., 2014). Mosquito DNVs are generally lethal to young larvae but are tolerated by older larvae, which develop into infected adults that complete the pathogen life cycle by inoculating virus into the aquatic larval environment (Carlson, Suchman & Buchatsky, 2006). Unlike the Aedes aegypti densovirus (AeDNV), which is generally lethal to young larvae and virulent to adults in a dose-dependent manner (Ledermann et al., 2004), the An. gambiae densovirus (AgDNV) has minimal pathogenic effects in larvae (Ren, Hoiczyk & Rasgon, 2008). In contrast to AeDNV, AgDNV does not replicate substantially in the immature or early adult life stages of An. gambiae, perhaps explaining minimal larval virulence. Instead, AgDNV replicates preferentially in adult mosquitoes (Ren & Rasgon, 2010).

To further evaluate the suitability of AgDNV for use in a paratransgenic malaria control strategy we studied the effect of infection on adult An. gambiae mosquitoes. By examining life history traits and the transcriptomic response of Anopheles mosquitoes to AgDNV infection we found minimal impact of AgDNV upon the mosquito host at the molecular level or upon adult fitness. Taken together, these data suggest that AgDNV could be a useful agent for paratransgenesis in An. gambiae mosquitoes as there is minimal fitness or transcriptome impact on the host after infection.

Materials and Methods

Cell culture, mosquito infection and rearing conditions

Sua5B cells, which are naturally infected with AgDNV (Ren, Hoiczyk & Rasgon, 2008), were passaged weekly in Schneider’s media with 10% FBS. AgDNV was obtained from the infected cell line Sua5B and first-instar mosquito larvae infected by exposure to purified virus as previously described (Ren, Hoiczyk & Rasgon, 2008). Control mosquitoes were mock infected with Schneider’s medium. After infection, mosquitoes were reared in 2L pans according to a standard feeding protocol as described (Ren & Rasgon, 2010).

Life-table analysis

At the pupal stage, cups of emerging pupae were placed in cages and removed 12 h later ensuring that adults were of similar ages. Mosquitoes were allowed access to 10% sucrose but were not blood fed. Mortality was accessed daily at the same time ±1 h. There were 3–4 cages per treatment (approximately 50 mosquitoes per cage), and the entire experiment was replicated three times (total = 740 AgDNV-infected mosquitoes, 860 control mosquitoes). For AgDNV treatments, collected dead mosquitoes were assayed for AgDNV infection by PCR amplification of a ∼300 bp fragment of the AgDNV capsid gene as described (Ren, Hoiczyk & Rasgon, 2008); infection rates per cage ranged from 87% to 100%. The experiment included both males and females, but mosquitoes were not sexed for analysis. Data were analyzed by the Gehan–Breslow–Wilcoxon test using GraphPad Prism 5.

RNA extraction and microarrays

Affymetrix GeneChip microarrays were used to assess the effect of AgDNV infection on An. gambiae gene transcription. First instar larvae were infected or mock infected as described above and reared to adulthood. At 10 days post-emergence when AgDNV titers are at their highest (Ren & Rasgon, 2010) mosquitoes were processed for analysis. For each biological replicate, pools of 20 adult mosquitoes were processed (mosquitoes were not sexed). There were three replicates per treatment. 20 randomly selected mosquitoes per replicate were tested by PCR (Ren, Hoiczyk & Rasgon, 2008) to confirm AgDNV infection; 100% of mosquitoes treated with virus as larvae were positive for infection by PCR compared to 0% of control mosquitoes. Mosquitoes were homogenized and lysed with Lysing Matrix D (MP bio) in 1 ml of Trizol reagent (Invitrogen) by rapid agitation in a FastPrep 120 Instrument (MP Biomedicals, Solon, OH) for 45 s at speed setting 6 and placed on ice for 2 min. Homogenization and ice incubation was repeated twice or until the samples were completely homogenized. After homogenization, RNA was extracted, purified and quantified as previously described (Hughes et al., 2011b). For each array 100 ng total RNA was hybridized to the Affymetrix Plasmodium/Anopheles microarray using the Affymetrix 3′ IVT express kit, according to manufacturer’s recommended protocol. Hybridization cocktails were prepared as recommended for arrays of Standard format using reagents in the Affymetrix Hybridization, Wash, and Stain kit. Hybridization was performed at 45 °C for 16 h at 60 rpm in the Affymetrix rotisserie hybridization oven. The signal amplification protocol for washing and staining of eukaryotic targets was performed in an automated fluidics station (Affymetrix FS450) using Affymetrix protocol FS450_0004. Arrays were transferred to a GCS3000 laser scanner with autoloader and 3G upgrade (Affymetrix) and scanned at an emission wavelength of 570 nm at 2.5 µm resolution. Quality assessment of hybridizations and scans was performed with Expression Console software. Detailed analyses were performed using Partek Genomics Suite version 6.4. GC-RMA algorithm defaults were used for background correction, normalization and summarization of probesets. Analysis of variance (AVOVA) was performed with linear contrasts for each densovirus treatment vs. control. Gene lists were developed based on 1.75 or 2.0 fold change (FC) or greater gene expression using a false discovery rate of P < 0.05. Raw Affymetrix CEL files are available as Supplemental data at http://rasgonlab.files.wordpress.com/2014/08/renetalrawaffymetrixdata.zip (controls: W1.CEL, W2.CEL, W3.CEL; treatment: AgDNV1.CEL, AgDNV2.CEL, AgDNV3.CEL).

qPCR validation of microarray analysis

qPCR analysis was completed as previously described (Hughes et al., 2011b). Briefly, RNA was extracted from pools of ten mosquitoes either infected with densovirus or uninfected using an RNeasy mini kit (Qiagen). Five biological replicates were completed for each treatment. RNA was DNase treated (Ambion) and cDNA synthesized using superscript III (Invitrogen) following manufactures guidelines. qPCR was completed using a Rotor gene Q (Qiagen) using Rotor gene SYBR green PCR kit (Qiagen) according to manufactures guidelines. To validate microarray results we assayed five genes selected both from genes affected and not affected in the microarray analysis (AMMECR1, Rel1, Rel2, cactus and caspar; primers in Table 1). Expression of each target gene was normalizing to expression of the ribosomal protein S7 gene (Table 1). All qPCRs were performed in triplicate. Melt curve analysis was completed on all PCRs. Determinations of relative expression were calculated using qGENE (Joehanes & Nelson, 2008).

Table 1 PCR and qPCR primer sequences.

Primers used for qPCR validation of microarray results and detection of AgDNV.

Gene name	AGAP/GenBank ID#	Primers (5′–3′)	
Rel1	AGAP009515	Forward: TCAACAGATGCCAAAAGAGGAAAT	
		Reverse: CTGGTTGGAGGGATTGTG	
Caspar	AGAP006473	Forward: TCCACACATGCAACCTGTTT	
		Reverse: CTCGCTGCAGCACAGCGGTA	
Rel2	AGAP006747	Forward: CGGTGCTCCTCGTAATGACT	
		Reverse: GTATCGTTGCGTCGGATTG	
Cactus	AGAP007938	Forward: GAACGGCTGCGCTTTAACA	
		Reverse: TCGTTCAAGTTCTGTGCAAGTGT	
AMMECR1	AGAP000328	Forward: AAGAGACTCCCGTTTCTCGCCAAT	
		Reverse: TCGAGCCACGCTCATTGTAGAACT	
S7	AGAP010592	Forward: CATTCTGCCCAAACCGATG	
		Reverse: AACGCGGTCTCTTCTGCTTG	
AgDNV capsid gene	EU233812	Forward: CAGAAGGATCAGGTGCAG	
		Reverse: GTTACTCCAAGAGCTACTC	

Results and Discussion

Life table analysis

We used life table analysis to assess the fitness effects of AgDNV infection on adult An. gambiae (Keele strain) mosquitoes. For replicates 1 and 2, survival trajectories of AgDNV infected and uninfected mosquitoes were not significantly different (Table 2, Figs. 1A and 1B). For replicate 3, mosquitoes in both treatments had significantly elevated mortality compared to replicates 1 and 2, and AgDNV-infected mosquitoes had a slight, but statistically significant reduction in lifespan (P = 0.003) (Table 2, Fig. 1C). We do not know why results from replicate 3 differed from replicates 1 and 2, but the significant reduction (greater than 50%) in lifespan for both treatments in replicate 3 suggests other confounding factors besides AgDNV infection are at play.

Figure 1 Survival of AgDNV infected or uninfected adult Anopheles gambiae.

Mosquitoes infected at the larvae stage with AgDNV purified from Sua5B cells (or mock infection as control; WT) were assessed for adult survivorship every 24 h until all the mosquitoes had perished. (A), (B) and (C) refer to replicates 1, 2 and 3 respectively.

Table 2 Survival statistics for individual experimental replicates.

Replicate	Treatment	N	Mean lifespan (days)	Chi-square	P value	
1	WT	344	24	0.047	0.828	
	AgDNV	289	25			
2	WT	298	23	2.51	0.113	
	AgDNV	199	22			
3	WT	218	11	8.92	0.003	
	AgDNV	252	9			

Microarray analysis

To evaluate the effect of AgDNV infection on host gene expression, we completed microarray analysis comparing infected versus uninfected adult mosquitoes. AgDNV infection had very limited effect on An. gambiae gene expression with global expression analysis identifying only 4 genes modestly differentially up-regulated (fold-change (FC) ≥ 2, P < 0.05) in response to infection, and no genes significantly down-regulated (Table 3). qPCR of selected genes showed similar results (R2 = 0.84, P = 0.03; Fig. 2). These results are in stark contrast to the effect of the human pathogen O’Nyong-nyong virus in An. gambiae, where infection resulted in the regulation of 253 genes (152 up-regulated, 102 down-regulated), including many genes involved in anti-viral and innate immune pathways (Waldock, Olson & Christophides, 2012). However, our results are similar to other studies examining gene expression in response to DNV infection in other insects. Minimal effects on host gene expression were seen in the moth Spodoptera frugiperda with the up-regulation of only 8 genes after injection of DNV into fat body tissue (Barat-Houari et al., 2006). Using subtractive hybridization to identify differentially expressed genes in Bombyx mori in response to virus infection, only 28 genes were found in a moth line resistant to DNV, while infection in susceptible line lead to differential expression of 23 genes (Bao et al., 2008). While infection with other microbes profoundly affects gene expression in An. gambiae (Abrantes et al., 2008), AgDNV (and DNVs in general) seem to neither strongly elicit nor suppress global gene expression patterns. Further studies are required to ascertain whether the virus employs mechanism(s) to avoid modulating the host transcriptome.

Figure 2 qPCR validation of microarray data.

Validation of microarray data in DNV-infected adult An. gambiae mosquitoes. Log2 fold change values for both microarray and qPCR methods were compared for 5 selected An. gambiae genes (AMMECR1, Rel1, Rel2, cactus and caspar). P = 0.03.

Table 3 Significantly regulated (P < 0.05, FC >1.75) genes in AgDNV-infected An. gambiae mosquitoes identified by microarray analysis.

Affymetrix probe	AGAP number	Gene	Function	P value	Fold-change	
Ag.3R.866.0.CDS_at	AGAP008277	TRYPSINOGEN 2	Peptide bond hydrolysis	0.0148	2.40	
Ag.X.430.0_CDS_at	AGAP000328	AMME SYNDROME CANDIDATE GENE 1	Unknown	0.0093	2.21	
Ag.X.341_CDS_a_at	AGAP001039	CYTOCHROME P450	Redox	0.0327	2.20	
Ag.3R.900.4_s_at	AGAP009368	PLUGIN	Mating plug	0.0400	2.00	
Ag.2L.2922.0_a_at	AGAP007116	CONSERVED HYPOTHETICAL PROTEIN	Unknown	0.0399	1.99	
Ag.2L.1432.0_CDS_a_at	AGAP006418	VENOM ALLERGIN	Secreted	0.0439	1.80	
Ag.3R.292.0_CDS_a_at	AGAP009036	SODIUM/HYDROGEN EXCHANGER CELLULAR
RETINALDEHYDE BINDING	Ion transport	0.0171	1.75	
Ag.3R.1298.0_CDS_s_at	AGAP009365	CRALBP	Binding	0.0431	1.66	
Ag.3L.707.0_UTR_at	AGAP010732	ZINC FINGER PROTEIN	Binding	0.0414	1.66	
Ag.2R.861.0_CDS_at	AGAP001281, AGAP001282	INWARD RECTIFIER POTASSIUM
CHANNEL SPECTRIN REPEAT SYNAPTIC NUCLEAR	Ion transport	0.0384	−1.63	
Ag.UNKN.1513.0_CDS_s_at	AGAP009554	ENVELOPE SERINE/THREONINE-PROTEIN	Actin binding	0.0228	−1.65	
Ag.2R.3544.0_CDS_at	AGAP004096	PHOSPHATASE 2A SUBUNIT EPSILON	Binding	0.0432	−1.72	
Ag.UNKN.2468.0_s_at	?	Unknown	Unknown	0.0012	−1.73	
Ag.UNKN.2158.0_CDS_at	?	Unknown
GLUCOSE DEHYDROGENASE	Unknown	0.0290	−1.78	
Ag.2R.613.1_CDS_s_at	AGAP003785	[ACCEPTOR} PRECURSOR	Dehydrogenase activity	0.0049	−1.79	

Of the 4 genes marginally up-regulated by AgDNV infection, two are putatively associated with stress response (cytochrome P450 (CYP450) and AMME syndrome candidate gene 1 (AMMECR1)). In mosquitoes, CYP450s are detoxification enzymes expressed when the insect is under oxidative stress (Feyereisen, 1999) and are known to be expressed when the mosquito is infected by pathogens (Abrantes et al., 2008). No functional studies have been conducted on AMMERCR1 in insects. The other two identified significantly regulated genes are trypsinogen 2 and plugin. Trypsinogen is the precursor of trypsin, which is involved in hydrolysis of peptide bonds during bloodmeal digestion (Noriega et al., 1996) while plugin is one of the major components of the mating plug (Le et al., 2013).

When we applied a less stringent FC criteria (1.75 FC), 15 genes were differently regulated in response to DNV virus infection (Table 3) with 9 genes up-regulated and 6 down-regulated. Besides functionally unknown genes, the rest of the genes are classified to transport-, metabolism- or binding-related transcripts (Table 3).

Paratransgenesis is an approach that attempts to modulate vector competence of the vector by manipulation of microorganisms within the host (Beard, Cordon-Rosales & Durvasula, 2002; Riehle & Jacobs-Lorena, 2005). Malaria researchers have focused on paratransgenesis as a novel alternative to traditional transgenic strategies (Favia et al., 2007). Our results suggest that AgDNV infection has minimal impact on survival or gene expression of its mosquito host making it a potentially attractive agent for paratransgenesis in An. gambiae. It should be noted that survival is only one component of the complex amalgam of traits that, collectively make up “fitness” and that further studies need to be performed to assess the effect of AgDNV infection on other fitness components (such as development time, fecundity, and mating behavior).

We thank the Johns Hopkins Malaria Research Institute Gene Array Core Facility (JHMRI-GACF) for assistance with microarrays.

Additional Information and Declarations

Competing Interests

Author Contributions

Microarray Data Deposition

Xiaoxia Ren is an employee of Pharmaceutics International Inc; all research was completed prior to onset of employment.

Xiaoxia Ren conceived and designed the experiments, performed the experiments, analyzed the data, contributed reagents/materials/analysis tools, wrote the paper, prepared figures and/or tables.

Grant L. Hughes conceived and designed the experiments, performed the experiments, analyzed the data, wrote the paper, prepared figures and/or tables, reviewed drafts of the paper.

Guodong Niu performed the experiments, prepared figures and/or tables.

Yasutsugu Suzuki performed the experiments, reviewed drafts of the paper.

Jason L. Rasgon conceived and designed the experiments, analyzed the data, contributed reagents/materials/analysis tools, wrote the paper, prepared figures and/or tables, reviewed drafts of the paper.

The following information was supplied regarding the deposition of microarray data:

Raw Affymetrix data files are available at:

http://rasgonlab.files.wordpress.com/2014/08/renetalrawaffymetrixdata.zip.

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
