# Peer review of "Anopheles gambiae densovirus (AgDNV) has negligible effects on adult survival and transcriptome of its mosquito host"

_PeerJ, doi:10.7717/peerj.584_

## Round 0.1 · original submission · Minor Revisions

· Academic Editor

Minor Revisions

As you can see from the comments below, both reviewers found the manuscript to be well-written and the study, overall, to be well designed. That said, please address concerns by Reviewer #2 with regards to providing information about the methods (Experimental Design) and also rephrasing your interpretations of the data (whether or not you've actually shown that AgDNV is not highly virulent) or alternatively, including some of the recommended experiments to supplement the current data.

Reviewer 1 ·

Basic reporting

No comments, this is a case in which the article meets the journal standard

Experimental design

The experimental design meets the journal standard

Validity of the findings

No comments

Comments for the author

The article deals with a very interesting subject and the results presented are original and coherent with the experimental design. I enjoyed reading it.
Even though sexing of the pre-adult mosquitoes as well as a more stringent explanation of why replicate 3 differed from the other two in the life table analysis, would have improved the overall quality of the manuscript, I think it disserves publication in PeerJ. In fact the final findings contribute to a better understanding on the relationships between Densoviruses and Insects and provide more information to set-up paratransgenic approach to contrast vector borne diseases.

I suggest only a couple of “stylistic” minor modifications:
1) Introduction, lane 32, “arthropod borne diseases instead of arthropod borne disease.
2) Results and Discussion, last paragraph starting from lane 167. It replicates what already pinpointed in the introduction and I would leave just the last sentence. Alternatively it should be done a better link between this paragraph and the previous one.

Reviewer 2 ·

Basic reporting

This study explores the virulence of densovirus AgDNV in the adult stage of the mosquito, Anopheles gambiae. Based on survival studies as well as microarray analyses, the authors conclude that AgDNV is not highly virulent in A. gambiae. The paper is well written and adheres to all basic reporting guidelines.

Experimental design

The experiments are well designed. As for specifics aspects, please include the gene used as the reference for relative quantification by RT-PCR (and include the primers in table 1), or describe this method in greater detail. Also, include in table 1 the AGAP IDs of the genes assayed by quantitative PCR.

Validity of the findings

The data are supportive of the hypothesis of this study. Uninfected and infected adult mosquitoes have similar lifespans and, at the one timepoint assayed, infection does not affect adult mosquito gene expression. However, although the data are supportive, the global conclusion that “Anopheles gambiae densovirus (AgDNV) is not highly virulent in its mosquito host” seems a bit premature. The study that the authors cite to state that AgDNV has minimal pathogenic effects in larvae only looked at survival. Other important effects of virulence were not measured, such as the eclosion rate, the time to eclosion and adult mosquito size. In the present study mosquito survival was measured, but the equally important trait of fecundity was not measured. Gene expression was only measured at one timepoint, and virus-induced behavioral changes were not assessed (e.g., susceptibility to predation). Granted, this last trait would be difficult to measure, but the point is that although the data presented here are supportive of the hypothesis that AgDNV is not highly virulent in A. gambiae, the assertive conclusions that AgDNV is “not highly virulent” or presents a “lack of strong fitness cost” are premature.

---

## Round 0.2 · accepted · Accept

· Academic Editor

Accept

Thank you for responding so quickly and thoroughly to these minor reviewer concerns. A couple of edits I would suggest include using "effects" in the title not "affects." Congratulations and best of luck in the future.